# Erdafitinib Inhibits Tumorigenesis of Human Lung Adenocarcinoma A549 by Inducing S-Phase Cell-Cycle Arrest as a CDK2 Inhibitor

**DOI:** 10.3390/molecules27196733

**Published:** 2022-10-09

**Authors:** Xinmin Meng, Xue Zhu, Jiali Ji, Hongqin Zhong, Xiyue Li, Hongqing Zhao, Guijuan Xie, Ke Wang, Hong Shu, Xun Wang

**Affiliations:** 1Department of Clinical Laboratory, Guangxi Medical University Cancer Hospital, Nanning 530021, China; 2National Health Commission (NHC) Key Laboratory of Nuclear Medicine, Jiangsu Key Laboratory of Molecular Nuclear Medicine, Jiangsu Institute of Nuclear Medicine, Wuxi 214063, China; 3Department of Radiopharmaceuticals, School of Pharmacy, Nanjing Medical University, Nanjing 210000, China; 4Department of Respiratory and Critical Care Medicine, The Affiliated Wuxi No.2 People’s Hospital of Nanjing Medical University, Wuxi 214002, China; 5Department of Respiratory and Critical Care Medicine, Wuxi Clinical College Affiliated to Nantong University, Wuxi 214002, China

**Keywords:** lung adenocarcinoma, erdafitinib, cell cycle, CDK2, E2F1-CDK1 signaling

## Abstract

Lung adenocarcinoma (LADC) is the most prevalent lung cancer sub-type, and targeted therapy developed in recent years has made progress in its treatment. Erdafitinib, a potent and selective pan-FGFR tyrosine kinase inhibitor, has been confirmed to be effective for the treatment of LADC; however, the molecular mechanism responsible for this effect is unclear. The in vitro study showed that erdafitinib exhibited an outstanding anti-cancer activity in human LADC cell line A549 by inducing S-phase cell-cycle arrest and cell apoptosis. The mechanistic study based on the transcriptomic data revealed that erdafitinib exerted its anti-cancer effect by affecting the cell cycle-related pathway, and CDK2 was the regulatory target of this drug. In addition, CDK2 overexpression significantly attenuated the anti-cancer effect of erdafitinib by affecting the transcriptional activity and expression of E2F1, as well as the expression of CDK1. The in vivo study showed that erdafitinib presented an obvious anti-cancer effect in the A549 xenograft mice model, which was accompanied by the reduced expression of CDK2. Thus, this study demonstrates the anti-cancer effect of erdafitinib against LADC for the first time based on in vitro and in vivo models, whose activity is achieved by targeting CDK2 and regulating downstream E2F1-CDK1 signaling. This study may be helpful for expanding the clinical application of erdafitinib in treating LADC.

## 1. Introduction

Lung cancer (LC) is currently the leading cause of cancer-related mortality in the world, with an incidence of 11.4% among all new cancer cases due to the GLOBOCAN 2020 [1,2,3]. Lung adenocarcinoma (LADC) is the most common type of LC, which accounts for around 40% of all cases [4,5]. At present, the treatment of LADC has been developed from traditional surgical treatment, radiotherapy and chemotherapy to more individualized and accurate treatment methods such as targeted treatment [6]. For example, EGFR gene mutations are detected in up to 50% of LADC patients; thus, highly efficacious epidermal growth factor receptor (EGFR) tyrosine kinase inhibitors (TKIs) have been developed and significantly improve the clinical outcome of these patients [7,8,9]. However, the lack of EGFR mutations or the acquired resistance of TKIs is the important challenge in the treatment of LC including LADC. Therefore, developing drugs that target other pathogenic genes is perhaps the effective strategy.

Erdafitinib (Balversa™) is developed by Janssen Pharmaceutical Companies as an orally administered, pan-fibroblast growth factor receptor (FGFR) inhibitor [10]. Erdafitinib is confirmed to be effective for the treatment of cancers with amplifications, mutations and fusions of FGFRs [11], and it is now approved for treating patients with urothelial cancer whose tumors harbor FGFR2 or FGFR3 mutations or fusions [12]. Erdafitinib is also being investigated as a treatment for other cancers including breast cancer, liver cancer, lung cancer, prostate cancer and esophageal cancer [13]. Chandrani et al. have reported that FGFR mutation represents an opportunity for targeted therapy in LADC, and FGFR inhibitors may be extended to evaluate in patients with FGFR-alteration LADC [14]. Urrutia et al. have reported the first case of a response to erdafitinib in a patient with stage IV adenocarcinoma of the lung [15]. In this study, the cytotoxic effect of erdafitinib against LADC was first investigated using in vitro and in vivo models, and the detailed mechanism responsible for erdafitinib’s effect as well as the potential target were further explored.

## 2. Results

### 2.1. Erdafitinib Induces S-Phase Cell-Cycle Arrest and Cell Apoptosis in Human Lung Adenocarcinoma A549 Cells

The effect of erdafitinib on cell viability was first examined by a MTT assay in LADC cells. As shown in Figure 1A,B, erdafitinib showed obviously cytotoxic effects in human lung adenocarcinoma cell lines including A549, H1975, H2009, Calu-3 and PC-9. Then, A549 cells with FGFR proteins (FGFR 1–3) overexpression (Appendix A) were used for the subsequent experiments. Erdafitinib significantly inhibited the cell growth of A549 cells in a dose- and time-dependent manner (IC_50_ at 24 h was 7.76 μΜ), while it showed a slight effect on human embryonic lung fibroblast cell line MRC-5 (Figure 1C). Then, the effects of erdafitinib on cell cycle and cell apoptosis were examined by flow cytometry in A549 cells. As shown in Figure 1D,E, erdafitinib significantly induced S-phase cell-cycle arrest and the cell apoptosis of A549 cells in a dose-dependent manner.

### 2.2. Transcriptomic Analysis of Human Lung Adenocarcinoma A549 Cells upon Erdafitinib Treatment

Transcriptome sequencing was used to systematically evaluate the mechanism responsible for erdafitinib. As shown in Figure 2A, a total of 2573 differentially expressed genes (DEGs) were identified with cut-off values of fragments per kilobase of transcript per million mapped reads: (FPKM) > 1, absolute log2 FC > 1, and Padj < 0.05 in erdafitinib (10 μM, 24 h)-treated A549 cells compared to control (erdafitinib, 0 μM, 24 h). Among these, 1401 genes were up-regulated, and 1172 genes were down-regulated. Then, a GO functional analysis was performed on 2573 DEGs and the results show that the large proportion of enriched GO terms were related to cell cycle (Figure 2B). Finally, a KEGG pathways analysis was performed on 2573 DEGs and the results show that the changed genes were enriched in cell cycle, DNA replication, p53 signaling pathway and cellular senescence (Figure 2C). A PPI analysis was conducted by STRING website and Cytoscape 3.7.2 software. The core network related to cell cycle was drawn and CDK2 presented as the key protein regulated by erdafitinib (10 μM, 24 h) (Figure 2D). Based on the transcriptomic data, the cell cycle-related pathway was further analyzed.

### 2.3. Cell-Cycle Network Analysis of Human Lung Adenocarcinoma A549 Cells upon Erdafitinib Treatment

The cell-cycle pathway was most affected by erdafitinib in A549 cells, then the change of cell-cycle network upon erdafitinib treatment was further investigated. As shown in Figure 3A, 46 cell cycle-related genes were dysregulated and presented as a GSEA heat map. As shown in Figure 3B, a PPI network of 46 cell cycle-related proteins was constructed using the STRING database (score > 0.9). As shown in Figure 3C, the associations of CDK2-related proteins were analyzed by the PPI network. In the PPI analysis, CDK1, E2F1 and CDK2 seemed to play the important roles in the effect of erdafitinib. Then, the mRNA and protein expressions of CDK1, E2F1 and CDK2 in A549 cells upon erdafitinib treatment (10 μM for 24 h) were confirmed to be down-regulated by qRT-PCR and Western blot analysis, which were consistent with the results of RNA-seq (Figure 3D,E and Appendix A).

### 2.4. CDK2 Is the Target of Erdafitinib in Human Lung Adenocarcinoma A549 Cells

To further analyze the involvement of the CDK2-E2F1-CDK1 signaling axis in the erdafitinib-induced S-phase cell-cycle arrest, CDK2 overexpression was conducted in A549 cells. As shown in Figure 4A–D, CDK2 overexpression significantly reduced cytotoxicity upon erdafitinib treatment, which presented as an attenuation of S-phase cell-cycle arrest and cell apoptosis. As shown Figure 4E and Appendix A, CDK2 overexpression significantly attenuated the down-regulated effect of erdafitinib on E2F1 and CDK1 expressions. Then, the association of CDK2, E2F1 and CDK1 was evaluated using a dual-luciferase reporter assay; the results show that CDK2 could affect the expression and transcriptional activity of E2F1, and E2F1 was the transcription factor of CDK1. The data indicated that CDK2 was the target of erdafitinib in A549 cells.

### 2.5. Erdafitinib Suppresses Tumor Growth in a A549 Xenograft Mouse Model

To further confirm the in vivo effect of erdafitinib, the A549 xenograft mouse model was established. Nude mice were intraperitoneally injected with vehicle or erdafitinib (10 mg/kg/day) for 21 days. As shown in Figure 5A,B, erdafitinib significantly inhibited tumor growth and reduced tumor volume. In addition, the resected tumor tissues were stained with H&E, Ki67, TUNEL and CDK2. H&E staining showed that erdafitinib treatment induced morphological changes with the signs of cell necrosis and infiltration of inflammatory cells. Then, erdafitinib treatment dramatically decreased Ki67-positive cells (indicative of cell proliferation ability) and increased TUNEL-positive cells (indicative of cell apoptosis), as well as reducing the expression of CDK2 in vivo (Figure 5C).

## 3. Discussion

Erdafitinib is a tyrosine kinase inhibitor (TKI) of fibroblast growth factor receptor (FGFR), which has been approved to treat locally advanced or metastatic urothelial carcinoma by the FDA [16]. Erdafitinib (chemical name: *N*-(3,5 dimethoxyphenyl)-*N*′-(1-methylethyl)-*N*-[3-(1-methyl-1H-pyrazol-4-yl) quinoxalin-6-yl] ethane-1,2 diamine, molecular formula C_25_H_30_N_6_O_2_) binds to an inactive DGF-Din conformation of FGFR1 and is classified as a type I½ inhibitor [17]. Recently, clinical or laboratory data have revealed that erdafitinib shows a therapeutic effect against human lung adenocarcinoma; however, knowledge about the specific mechanism of this drug is still lacking. In this study, in vitro and in vivo models using human lung adenocarcinoma A549 cells were constructed for evaluating the effect and underlying mechanism of erdafitinib. Our preliminary experiments revealed that FGFR1 was highly expressed in A549 cells and down-regulated by erdafitinib treatment. Then, further investigation on the mechanism responsible for erdafitinib revealed that cell cycle-related signaling was the main target of this drug based on transcriptomic data.

As a complex sequence of events through which a cell duplicates its contents and divides, abnormal activity of the cell cycle represents a driving force of tumorigenesis [18,19,20]. Recently, cell-cycle regulation has become an effective anti-tumor strategy [21,22,23,24]. Cyclin-dependent kinases (CDKs), the key regulatory enzymes, are involved in cell-cycle checkpoints regulation [25,26]. The CDK family is known to regulate the cell cycle, transcription and splicing, and deregulation of any of the stages of the cell cycle or transcription leading to apoptosis [26]. Cyclins concentration changes periodically throughout the cell cycle, and a PSTAIRE motif allows cyclins to form dimer complexes with corresponding CDKs, enabling a conformational change of the residues that are responsible for ATP binding. Following the binding of the cyclin to CDK, a small L12 helix at the primary sequence of the T-loop is transformed into a beta-strand, causing the active site and T-loop to be reoriented [27]. Drug discovery targeting CDK4/6 becomes a hotspot with the clinical success of CDK4/6 inhibitors, which have been approved for the treatment of hormone receptor-positive breast cancer [28,29,30,31]. In addition, inhibitors targeting other cell-cycle CDKs are currently in clinical trials [32]. In this study, transcriptome sequencing and analysis revealed that erdafitinib mainly affected the cell viability of A549 cells by inducing S-phage cell-cycle arrest. The PPI network analysis found that CDK2 was the main regulatory target of this drug. To further investigate the role of CDK2 in erdafitinib’s effect, CDK2 overexpression was conducted because it is the upstream of CDK1 [33]. The results show that CDK2 overexpression significantly attenuated the effect of erdafitinib on S-phage cell-cycle arrest and cell apoptosis. Then, the dual-luciferase reporter assay and Western blot analysis showed that CDK2 could affect the transcriptional activity and expression of E2F1, and subsequently affect the expression of CDK1. In addition, p21, as a known CDK inhibitor that can bind CDK2 and inhibit CDK2 activity [34], showed no significant change in mRNA expression based on the transcriptomic results in A549 cells upon erdafitinib treatment. Further, the Western blotting results show that the protein expression of p21 was slightly increased after erdafitinib treatment; however, there was no significant difference (data not shown). Our data indicated that erdafitinib regulated E2F1-CDK1 signaling by affecting CDK2 expression; however, whether the drug enters the cancer cell to exert its effect needs to be further studied. Furthermore, the in vivo experiments using the A549 xenograft mouse model were consistent with the data from the in vitro experiments. In addition, the in vivo data showed that an intraperitoneal injection of erdafitinib (10 mg/kg/day) in A549 xenograft mice for 21 days was more effective than erdafitinib (5 mg/kg/day) (Appendix A).

## 4. Materials and Methods

### 4.1. Chemicals and Reagents

Erdafitinib (purity > 99%) was obtained from Qianyan Biotech (Shanghai, China) and dissolved in dimethyl sulfoxide (DMSO). Chemicals were obtained from Sigma-Aldrich (St. Louis, MO, USA) and MCE (Shanghai, China). Reagents were obtained from Thermo Fisher (Rockford, IL, USA), Santa Cruz Biotechnology (Dallas, CA, USA) and Abcam (Cambridge, MA, USA). Other materials were obtained from Sangon (Shanghai, China) and Beyotime (Nantong, China).

### 4.2. Cell Lines and Culture

The human lung adenocarcinoma A549, H1975, H2009, Calu-3 and PC-9 cells as well as human embryonic lung fibroblast MRC5 cells were obtained from ATCC (American Type Culture Collection, Manassas, VA, USA) and SCBCAS (Shanghai Cell Bank of Chinese Academy of Sciences, Shanghai, China). The cells were maintained in DMEM (Dulbecco’s modified Eagle’s medium) with fetal bovine serum (FBS, 10%) and penicillin/streptomycin (P/S) and glutamine (2 mM) in a humidified atmosphere of 5% CO_2_ at 37 °C.

### 4.3. Cell Viability Analysis

MTT (3-(4,5-dimethylthiazol-2-yl)-2,5-diphenyltetrazolium bromide) assay was used for measuring cell viability [35]. The cells (1 × 10^4^ cells/well) were seeded in 96-well plates overnight. After the indicated treatment, 5 mg/mL of MTT solution was added for 4 h at 37 °C. The absorbance was determined with the fluorescence spectrophotometer at 490 nm (SpectraMax M5, Molecular Devices, San Jose, CA, USA). Cell viability was presented as a percentage of the value against the control.

### 4.4. Cell-Cycle Analysis

The cells were cultured in serum-free DMEM medium for 24 h to synchronize into the G0 phase and incubated with indicated treatments [36]. Cells with the indicated treatment were collected and fixed using ice-cold 70% ethanol. Then, propidium iodide (PI, 50 μg/mL) was added for incubation for 30 min in the dark. Cell-cycle distribution was determined using a FACSaria-I flow cytometer (Becton-Dickinson, Franklin Lakes, NJ, USA).

### 4.5. Cell Apoptosis Analysis

Cell apoptosis was assessed by double staining with Annexin V-FITC and PI, as previously described [37]. Cells with indicated treatment were harvested and fixed in 70% ethanol on ice for 30 min, and digested with 100 μg/mL of ribonuclease A for 20 min at 37 °C. Then, Annexin V-FITC and PI were added for staining for 20 min in the dark. The percentage of apoptotic cells was determined using a FACSaria-I flow cytometer (Becton-Dickinson, Franklin Lakes, NJ, USA).

### 4.6. RNA Sequencing and Analysis

RNA isolation, quantification and library preparation were conducted, and then RNA-seq was performed by Novogene Corporation (Tianjin, China) [38]. Differentially expressed genes (DEGs) referred to genes with a fold-change of >2.0 and an adjusted *p*-value of <0.05. A gene ontology (GO) analysis and KEGG pathways analysis were used for testing the statistical enrichment of DEGs.

### 4.7. Quantitative Real-Time PCR (qRT-PCR)

The extraction and purification of RNA were performed as mentioned above [39]. First-strand cDNA was synthesized from 2 μg of total RNA using the PrimeScript^TM^ RT-PCR kit (TaKaRa, Dalian, China) and quantified using SYBR Premix Ex Taq^TM^ (TaKaRa, Dalian, China) on an ABI 7500 Fast Real-Time system (ThermoFisher, Waltham, MA, USA). An endogenous control of GAPDH was used to normalize the data.

### 4.8. Western Blot Analysis

RIPA (radio-immunoprecipitation assay) lysis buffer was used for total protein extraction and the concentration of each sample was measured with a BCA protein assay kit [40]. SDS-PAGE at 10 or 15% was used to separate protein samples. Then, the proteins were transferred to PVDF membranes and the membranes were blocked and incubated with primary antibody at 4 °C overnight, followed by incubation with HRP (horseradish peroxidase)-conjugated secondary antibody at 37 °C for 2 h. The ECL (enhanced chemiluminescence) assay kit was used for visualization. An endogenous control of GAPDH was used to normalize each band.

### 4.9. Cell Transfection

The negative control pcDNA3.1-vector was obtained from Invitrogen (Carlsbad, CA, USA), and the CDK2 expression vector pcDNA3.1-CDK2 was obtained from GenePharma (Shanghai, China). The cells were transfected with empty or expression vector (1 μg/mL) using Lipofectamine 2000 reagent (Invitrogen, Carlsbad, CA, USA) at 37 °C [41]. Forty-eight hours later, the efficiency of transfection was determined by a Western blot analysis. Then, after transfection for 48 h, the cells could be used for the subsequent experiments.

### 4.10. Dual-Luciferase Reporter Assay

The effect of CDK2 on the transcriptional activity of E2F1 in HEK293T and A549 cells was determined by a dual-luciferase report assay using CDK1 promoter as substrate [42]. The human gene (CDK1) promoter was sub-cloned into pGL3 vector (Promega, Madison, WI, USA) in HEK293T or A549 cells. Co-transfection of gene promoter vector pcDNA3.1-E2F1 or pcDNA3.1-E2F1+pcDNA3.1-CDK2 was conducted, and a luciferase assay was performed 48 h after transfection using the Firefly/Renilla Dual-Luciferase Reporter Assay System (Promega, Madison, WI, USA).

### 4.11. Nude Mice Tumorigenesis Assay

Animals for the experiments were kept in a pathogen-free environment and fed adlib. A549-Luc cells (5 × 10^7^) were mixed with Matrigel (2:1) and injected subcutaneously into BALB/c nude mice (about 5-week-old, Changzhou Cavens, Changzhou, China). The mice were randomly divided into two groups (*n* = 3 per group): vehicle and erdafitinib (10 mg/kg), when the tumor volumes reached approximately 100 mm^3^. Body weight and tumor volumes were measured every other day. After treatment, the mice were intraperitoneally injected with D-luciferin working solution (15 mg/mL) and image analysis was performed using the Xenogen Living Image system (Alameda, CA, USA) [43]. Then, the tumors were removed, weighed and photographed. The animals’ care and use were approved by the Laboratory Animal Ethics Committee of Jiangsu Institute of Nuclear Medicine (JSINM-2022–007).

### 4.12. Histology and Immunohistochemistry

After treatment, the tumor tissues were embedded in paraffin and cut into 8 mm-thick sections. Then, H&E, TUNEL and Ki67 staining was performed using tumor sections [43]. Then, the expression of CDK2 was assessed using immunohistochemical staining. The DAB kit was used for visualization, and a light microscope (Olympus IX53, Tokyo, Japan) was used for recording the image.

### 4.13. Statistical Analysis

All the experiments were performed independently at least three times, and the data are expressed as the mean ± SD (standard deviation). All statistical analyses were carried out using SPSS 25.0 software. Statistical comparisons were conducted with the Student’s *t*-test between two groups and a one-way ANOVA followed by Tukey’s post hoc test among three groups. A *p*-value of < 0.05 was considered as statistically significant.

## 5. Conclusions

In summary, our laboratory data confirmed that erdafitinib was effective for the treatment of LADC, which worked by targeting FGFR1 and regulating CDK2 expression (Figure 6). A future clinical investigation of erdafitinib in LADC may provide a new approach for treating this type of cancer.

## Figures and Tables

**Figure 1 molecules-27-06733-f001:**
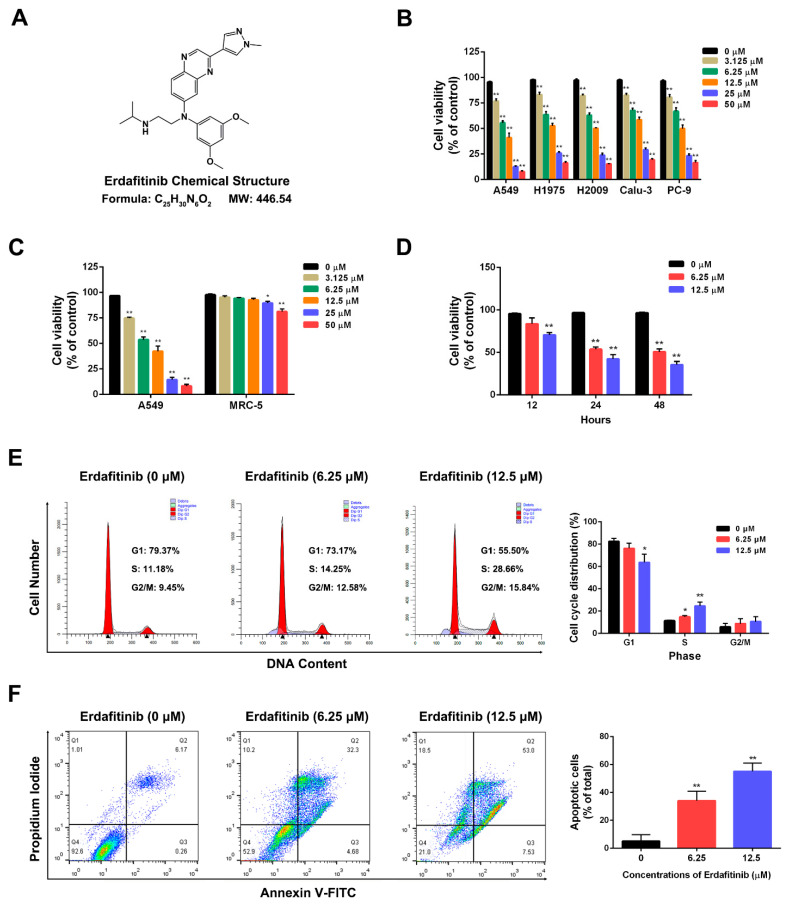
Erdafitinib inhibited cell viability of human lung adenocarcinoma A549 cells by inducing S-phage cell-cycle arrest and cell apoptosis. (**A**) The chemical structure of erdafitinib. (**B**) The effects of erdafitinib (0, 3.125, 6.25, 12.5, 25 and 50 μM, 24 h) on human lung adenocarcinoma cell lines including A549, H1975, H2009, Calu-3 and PC-9 were assessed by MTT assay. (**C**) The effects of erdafitinib (0, 3.125, 6.25, 12.5, 25 and 50 μM, 24 h) on human lung adenocarcinoma A549 cells and human embryonic lung fibroblast MRC-5 cells were assessed by MTT assay. (**D**) The effects of erdafitinib (0, 6.25, and 12.5 μM, 12, 24 and 48 h) on human lung adenocarcinoma A549 cells were assessed by MTT assay. (**E**) A549 cells were treated with erdafitinib (0, 6.25 and 12.5 μM, 24 h) and cell cycle was analyzed by flow cytometry and statistically analyzed. (**F**) A549 cells were treated with erdafitinib (0, 6.25 and 12.5 μM, 24 h) and cell apoptosis was analyzed by flow cytometry and statistically analyzed. The Annexin V+/PI- and Annexin V+/PI+ cells were considered as early and late apoptotic cells, respectively, and the sum of the above two was calculated as apoptotic cells. Data were represented as means ± SD (*n* = 3). * *p* < 0.05, ** *p* < 0.01 vs. erdafitinib (0 μM).

**Figure 2 molecules-27-06733-f002:**
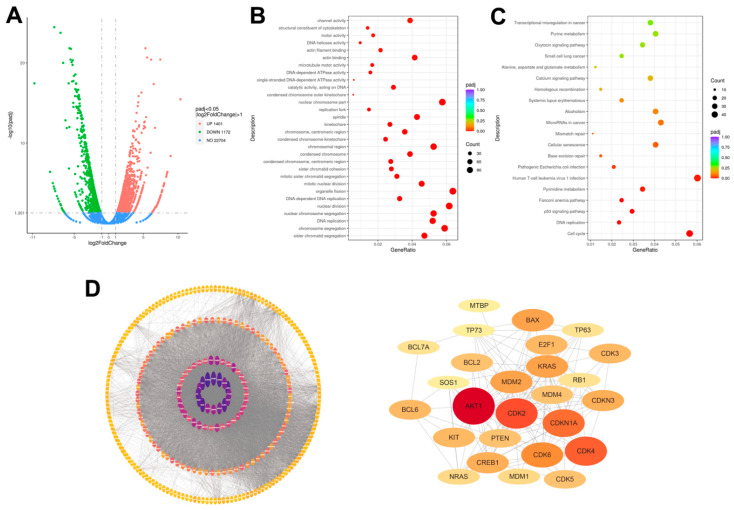
Transcriptomic analysis of human lung adenocarcinoma A549 cells upon erdafitinib treatment. A549 cells were treated with erdafitinib (10 μM) for 24 h, and RNA sequencing and analysis were conducted. (**A**) Volcano plots of DEGs (log2 fold change > 1 and Padj < 0.05). (**B**) GO analysis of DEGs. (**C**) KEGG analysis of DEGs. (**D**) PPI analysis of DEGs. The core network for five hundred significantly changed proteins (left panel) and the core network for cell injury-related proteins (right panel).

**Figure 3 molecules-27-06733-f003:**
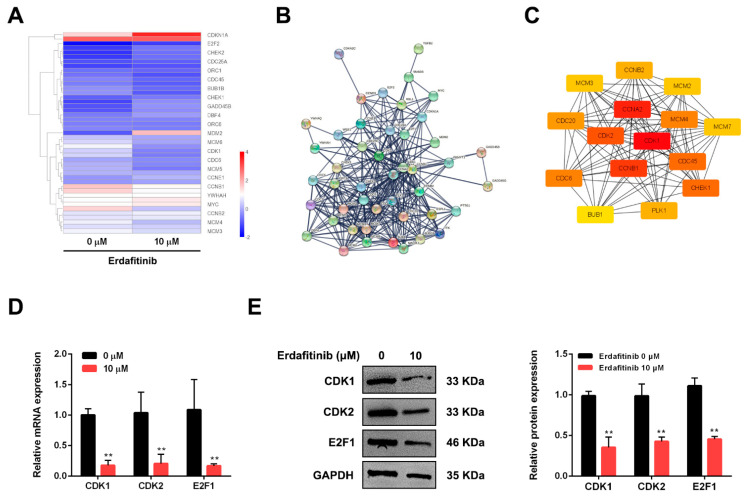
Cell-cycle network analysis of human lung adenocarcinoma A549 cells upon erdafitinib treatment. (**A**) Heatmap of DGEs related to cell cycle. (**B**) PPI network for the 46 important enzymes encoded by DEGs related to cell cycle. (**C**) PPI network for the CDK1/2 related proteins. (**D**) The mRNA expressions of CDK1, CDK2 and E2F1 were assessed by qRT-PCR in cells with or without erdafitinib treatment (10 μM, 24 h). (**E**) The protein expressions of CDK1, CDK2 and E2F1 were assessed by Western blot analysis in cells with or without erdafitinib treatment (10 μM, 24 h). Data were represented as means ± SD (*n* = 3). ** *p* < 0.01 vs. erdafitinib (0 μM).

**Figure 4 molecules-27-06733-f004:**
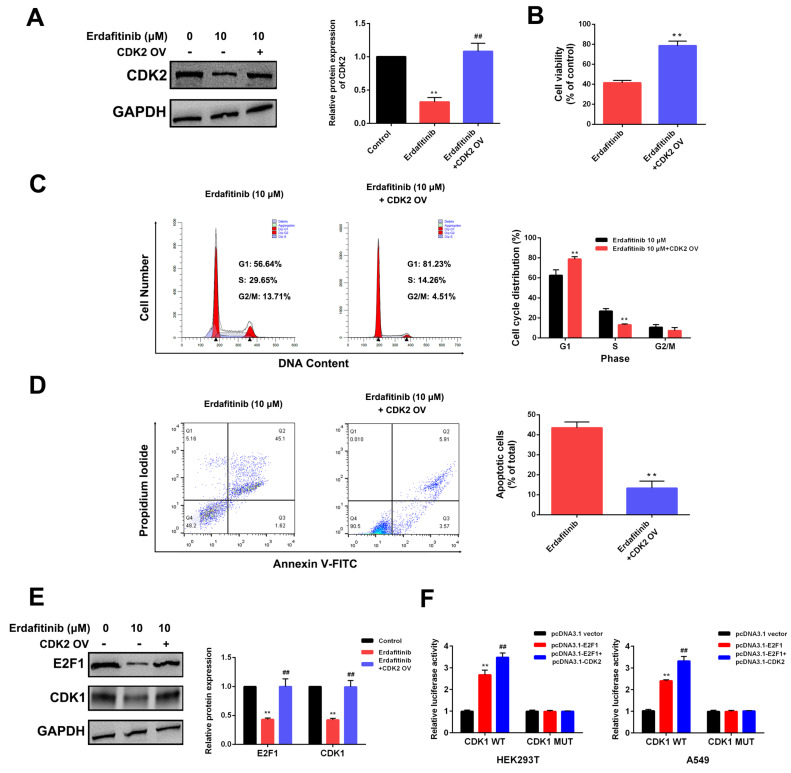
CDK2 overexpression attenuated erdafitinib’s cytotoxic effect on human lung adenocarcinoma A549 cells. A549 cells were transfected with pcDNA3.1-CDK2 for 48 h and then treated with erdafitinib (10 μM) for 24 h. (**A**) The expression of CDK2 was assessed by Western blot analysis. (**B**) Cell viability was assessed by MTT assay. (**C**) Cell cycle was assessed by flow cytometry and statistically analyzed. (**D**) Cell apoptosis was assessed by flow cytometry and statistically analyzed. The Annexin V+/PI- and Annexin V+/PI+ cells were considered as early and late apoptotic cells, respectively, and the sum of the above two was calculated as apoptotic cells. (**E**) The expressions of E2F1 and CDK1 were assessed by Western blot analysis. (**F**) The effect of CDK2 on the transcription activity of E2F1. The luciferase reporter vector containing WT CDK1 3′-UTR or MUT CDK1 3′-UTR and pcDNA3.1-vector, pcDNA3.1-E2F1 or pcDNA3.1-E2F1+pcDNA3.1-CDK2 were co-transfected into HEK293T and A549 cells, and luciferase activity was determined and normalized to Renilla luciferase. Data were represented as means ± SD (*n* = 3). ** *p* < 0.01 vs. control ^##^
*p* < 0.01 vs. erdafitinib.

**Figure 5 molecules-27-06733-f005:**
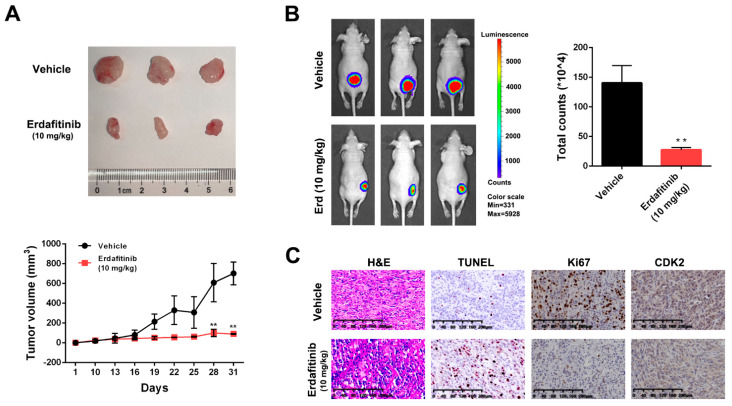
Erdafitinib suppresses tumor growth in a A549 xenograft mice model. (**A**) Tumor volumes were measured every other day. At the end of the treatment, the tumors were removed and photographed (*n* = 3). The images of the tumors are shown on the top panel and the tumor growth curve is displayed on the bottom panel. (**B**) Live imaging of the animals prior to euthanasia, and photon intensities are indicated to the right of the picture. (**C**) Representative images of hematoxylin-eosin (H&E), TUNEL, Ki67 and CDK2 staining in A549 xenografts. Data were expressed as mean ± SD of three experiments. ** *p* < 0.01 vs. vehicle.

**Figure 6 molecules-27-06733-f006:**
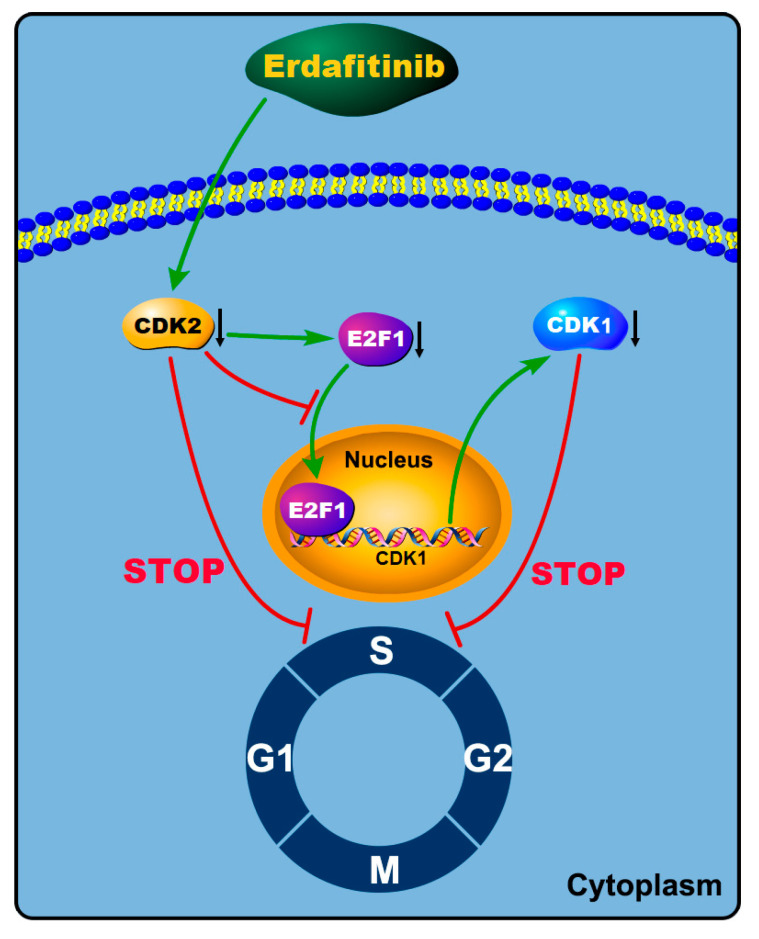
The proposed molecular mechanism of erdafitinib in inhibiting human lung adenocarcinoma A549 cells. Green arrow represents “promotion”. Red blocking symbol represents “inhibition”. Black down arrow represents down-regulation of protein expression.

## Data Availability

Data is contained within the article.

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
