# Peer review of "Erdafitinib Inhibits Tumorigenesis of Human Lung Adenocarcinoma A549 by Inducing S-Phase Cell-Cycle Arrest as a CDK2 Inhibitor"

_molecules, 2022, doi:10.3390/molecules27196733_

Round 1

Reviewer 1 Report

Journal: Molecules

Title: Erdafitinib Inhibits Tumorigenesis of Human Lung Adenocarcinoma A549 by Inducing S-phase Cell Cycle Arrest as A CDK2 Inhibitor

Major comments:

The manuscript is intriguing and novel in term of transcriptome study of the effect of TKI (Erdafitinib) on human lung A549 cancer cell’s gene and protein expression in both in vitro and in vivo models. The results pointed to CDK2-CDK1-E2F1 axis for the mechanistic pathway of apoptosis.

In Introduction:

The incidence of lung cancer should also refer to GLOBOCAN 2020.

In Methods:

The cell viability assay was to determine the absorbance of MTT dye. Please check the type of microplate reader whether it is a fluorescence microplate reader or not?? For statistical analysis, the one-way analysis is followed by which kind of statistical analysis to compare more than 2 groups?? This should be specified.

In Results:

The effect of Erdafitinib on human lung cancer is time-dependent or not. The variation of incubation time is required. The percentage of apoptotic cells is late apoptosis or early apoptosis or both, this should be specified in bar graphs or in the text. It should be early apoptotic or combined early and late apoptotic cells. Fig. 3D, 3E, Fig. 4A, 4E, there should be the dose response data of the Erdafitinib on the mRNA and protein levels of E2F1, CDK1 and CDK2. The results of qRT-PCR and Western blotting of 3 independent experiments should be provided in Suppl data. In Fig. 4E, there is a mark of ## without specified description in Figure legend whether it was compared with which bar graph?? The comparison of statistical significance value between bar graphs of Fig. 4E and 4 F are different, it should be consistent with each other or not?? The results of histology and immunohistochemistry should be described in details compared between vehicle and drug treatment group. The effects provided by the drug was dose response or not in in vivo model, the variation of the concentrations of the drug in xenograft model should be given.

In Discussion: In line 292, the statement that ……. because CDK2 is the upstream of CDK1, reference(s) should be provided. For the cell cycle control, the checkpoint proteins should be described in details. The authors should discuss more on the structure of Erdafitinib and the TK or FGFR. Re-depict the abstract figure (Fig.6) whether the drug can enter the cancer cell and show the cell compartments that were involved accordingly. The number of provided references are too small in number. The following papers may be included and considered in the Discussion part, PMID: 32005795, PMID: 30879017, DOI:10.3390/ijms21061960, https://doi.org/10.3390/cancers14020293, https://doi.org/10.1155/2021/6612365, https://doi.org/10.1186/s11658‑022‑00362‑4, PMID: 26345201, PMID: 29872310, etc.

Minor comments:

The references are updated; however, there were not so many references in the manuscript, of which there should be more in number. There are some typo and grammar errors that should be corrected by a native English speaker.

Author Response

Major comments:

The manuscript is intriguing and novel in term of transcriptome study of the effect of TKI (Erdafitinib) on human lung A549 cancer cell’s gene and protein expression in both in vitro and in vivo models. The results pointed to CDK2-CDK1-E2F1 axis for the mechanistic pathway of apoptosis.

In Introduction:

The incidence of lung cancer should also refer to GLOBOCAN 2020.

A: Thank you for your comments. Your comments are valuable. As you pointed out, the incidence of lung cancer was 11.4% among all new cancer cases due to the GLOBOCAN 2020, we have added it in the Introduction and cited the reference. See line 41-42.

In Methods:

(1) The cell viability assay was to determine the absorbance of MTT dye. Please check the type of microplate reader whether it is a fluorescence microplate reader or not??

A: Thank you for your comments. Your comments are valuable. In fact, the absorbance was determined with the fluorescence spectrophotometer at 490 nm (SpectraMax M5, Molecular Devices, CA, USA). This type of fluorescence microplate reader can also detect the absorbance of MTT dye at 490 nm or 570 nm, which is suitable for MTT assay.

(2) For statistical analysis, the one-way analysis is followed by which kind of statistical analysis to compare more than 2 groups?? This should be specified.

A: Thank you for your comments. Your comments are valuable. In fact, statistical comparisons were conducted with the Student’s t-test between two groups and one-way ANOVA followed by Tukey’s post hoc test among three groups. As you pointed out, we have revised it in the Methods. See line 155-157.

In Results:

(1) The effect of Erdafitinib on human lung cancer is time-dependent or not. The variation of incubation time is required.

A: Thank you for your comments. Your comments are valuable. As you pointed out, we investigated the effect of erdafitinib on A549 cells for 12, 24 and 48 h, respectively. Erdafitinib inhibited cell growth of A549 cells in a time-dependent manner; however, the effect was not significantly enhanced over 48 h. Relevant results have been supplemented in “Results” section, Fig. 1D. See line 165-166.

(2) The percentage of apoptotic cells is late apoptosis or early apoptosis or both, this should be specified in bar graphs or in the text. It should be early apoptotic or combined early and late apoptotic cells.

A: Thank you for your comments. Your comments are valuable. The Annexin V+/PI- and Annexin V+/PI+ cells were considered as early and late apoptotic cells, respectively, and the sum of the above two was calculated as apoptotic cells. As you pointed out, we have added it in the Figure legends. See line 182-183, 245-246.

(3) Fig. 3D, 3E, Fig. 4A, 4E, there should be the dose response data of the Erdafitinib on the mRNA and protein levels of E2F1, CDK1 and CDK2.

A: Thank you for your comments. Your comments are valuable. In fact, we conducted functional analysis using erdafitinib in a dose-dependent manner, and found the IC50 was 7.76 μΜ. Then, we conducted the RNA-seq analysis using 10 μΜ erdafitinib, which was little higher than IC50. To verify the RNA-seq results, the subsequent experiments should use 10 μΜ as the injured concentration, and more than 10 μΜ would be excessive damage and less than 10 μΜ would be less damage. So, we have not selected the other concentration.

(4) The results of qRT-PCR and Western blotting of 3 independent experiments should be provided in Suppl data.

A: Thank you for your comments. Your comments are valuable. As you pointed out, we have uploaded the suppl data about the results of qRT-PCR and Western blotting of 3 independent experiments.

(5) In Fig. 4E, there is a mark of ## without specified description in Figure legend whether it was compared with which bar graph?? The comparison of statistical significance value between bar graphs of Fig. 4E and 4 F are different, it should be consistent with each other or not??

A: Thank you for your comments. Your comments are valuable. As you pointed out, we have added the description of ## in Figure legend, in addition, we have revised the Figure 4F. See line 251-252.

(6) The results of histology and immunohistochemistry should be described in details compared between vehicle and drug treatment group.

A: Thank you for your comments. Your comments are valuable. As you pointed out, we have added the description of the results of histology and immunohistochemistry. See line 258-262.

(7) The effects provided by the drug was dose response or not in in vivo model, the variation of the concentrations of the drug in xenograft model should be given.

A: Thank you for your comments. Your comments are valuable. In fact, we selected the concentration in vivo based on the IC50 in vitro. We selected 5 mg/kg/day and 10 mg/kg/day of erdafitinib for the experiments, and found that intraperitoneal injection of erdafitinib (10 mg/kg/day) in A549 xenograft mice for 21 days was more effective than erdafitinib (5 mg/kg/day) (data not shown). So, we have not put the data in the Results, and we have added the discussion as you pointed out. See line 312-314.

In Discussion:

(1) In line 292, the statement that ……. because CDK2 is the upstream of CDK1, reference(s) should be provided.

A: Thank you for your comments. Your comments are valuable. As you pointed out, we have added the related reference (doi: 10.1038/sj.onc.1207446).

(2) For the cell cycle control, the checkpoint proteins should be described in details.

A: Thank you for your comments. Your comments are valuable. As you pointed out, we have added the related discussion. (CDK family is known to regulating the cell cycle, transcription and splicing, and deregulation of any of the stages of the cell cycle or transcription leads to apoptosis. Cyclins concentration changes periodically throughout the cell cycle, and a PSTAIRE motif allows cyclins to form dimer complexes with corresponding CDKs, enabling conformational change of residues responsible for ATP binding. Following binding of the cyclin to CDK, a small L12 helix at the primary sequence of the T-loop is transformed into a beta-strand, causing the active site and T-loop to be reoriented). See line 290-296.

(3) The authors should discuss more on the structure of Erdafitinib and the TK or FGFR.

A: Thank you for your comments. Your comments are valuable. As you pointed out, we have added the related discussion. (Erdafitinib (chemical name: N-(3,5    dimethoxyphenyl)-N’-(1-methylethyl)-N-[3-(1-methyl-1H-pyrazol-4-yl) quinoxalin-6-yl] ethane-1,2 diamine, molecular formula C25H30N6O2) binds to an inactive DGF-Din conformation of FGFR1 and is classified as a type I½ inhibitor). See line 274-277.

(4) Re-depict the abstract figure (Fig.6) whether the drug can enter the cancer cell and show the cell compartments that were involved accordingly.

A: Thank you for your comments. Erdafitinib regulates the expression of relative proteins, such as CDK1, CDK2 and E2F1, as well as the cell cycle. However, our current experimental results cannot determine the status of erdafitinib in cancer cells, and we discussed it in the Discussion. The cell compartments such as “Cytoplasm” and “Nucleus” have been added in Fig.6. See line 309-310.

(5) The number of provided references are too small in number. The following papers may be included and considered in the Discussion part, PMID: 32005795, PMID: 30879017, DOI:10.3390/ijms21061960, https://doi.org/10.3390/cancers14020293, https://doi.org/10.1155/2021/6612365, https://doi.org/10.1186/s11658022003624, PMID: 26345201, PMID: 29872310, etc.

A: Thank you for your comments. Your comments are valuable. As you pointed out, we have added the related references in the manuscript.

Minor comments:

The references are updated; however, there were not so many references in the manuscript, of which there should be more in number. There are some typo and grammar errors that should be corrected by a native English speaker.

A: Thank you for your comments. Your comments are valuable. As you pointed out, we have added the references and revised the typo and grammar errors according to a native English speaker.

Reviewer 2 Report

The manuscript entitled “Erdafitinib inhibits tumorigenesis of human lung adenocarcinoma A549 by inducing S-phase cell cycle arrest as a CDK2 inhibitor” shows a tyrosine kinase inhibitor, erdafitinib, induces S/G2 arrest in A549 lung adenocarcinoma cells, and suppresses tumorigenicity in xenograft model. This article is potentially interesting and may provide a novel option for treatment of lung cancer patients, however, this reviewer cannot recommend it with the present form.

(1)  Expression of internal CDK2 inhibitor, p21, should be checked.  

(2)  Figure 6

Indication arrows and bars for promotion/inhibition are confusing whether they are by phenomena (decreased expressions) or by the factors themselves.

(3)  Most of the experiments were performed using only one cell line, A549. This reviewer thinks that at least one more cell line should be checked for the key experiments (Figs. 4 and 5). Although this reviewer thinks that single cell study is not suitable for publication in journals of medical sciences, this reviewer would put this matter in the hands of the editor.

Author Response

The manuscript entitled “Erdafitinib inhibits tumorigenesis of human lung adenocarcinoma A549 by inducing S-phase cell cycle arrest as a CDK2 inhibitor” shows a tyrosine kinase inhibitor, erdafitinib, induces S/G2 arrest in A549 lung adenocarcinoma cells, and suppresses tumorigenicity in xenograft model. This article is potentially interesting and may provide a novel option for treatment of lung cancer patients, however, this reviewer cannot recommend it with the present form.

(1) Expression of internal CDK2 inhibitor, p21, should be checked. 

A: Thank you for your comments. Transcriptomic results show that there is no significant change in mRNA expression of p21 before and after erdafitinib treatment. Our western blotting results show that the protein expression of p21 is slightly increased after erdafitinib treatment, however, there is no significant difference. CDK2 overexpression did not affect protein expression of p21, so we have not put the data in the Results.

(2) Figure 6: Indication arrows and bars for promotion/inhibition are confusing whether they are by phenomena (decreased expressions) or by the factors themselves.

A: Thank you for your comments. Green arrow indicates promotion. Red blocking symbol indicates inhibition. Black down arrow represents down-regulation of protein expression. Detailed description has been supplemented in figure legend of Fig.6. See line 317-318.

(3)  Most of the experiments were performed using only one cell line, A549. This reviewer thinks that at least one more cell line should be checked for the key experiments (Figs. 4 and 5). Although this reviewer thinks that single cell study is not suitable for publication in journals of medical sciences, this reviewer would put this matter in the hands of the editor.

A: Thank you for your comments. Your comments are valuable. In fact, erdafitinib is a targeted drug which targets FGFR. As this, we select the human lung adenocarcinoma lines with FGFR1-3 overexpression for the experiments, and find that only A549 cells highly express FGFR1-3, which is suitable for the erdafitinib study. So, in this study, we use A549 cells for the all experiments. As you pointed out, we have added the description in the Results, and further will isolate and culture primary tumor cells with FGFR overexpression from LADC patients for the study. See line 164.

Round 2

Reviewer 1 Report

In each experiment there should be at least one reference included, please check and add the reference in all methods. For the Western blotting bands, some contains 2 lanes and some more than 2, there should be labeling above them to mark each lane what it represents for. For In Western blotting why did some lanes contain several bands and not consistent with other repeats.  It seems that the immunoblotting results are not consistent with each other. For the data (that were not shown) it would be better to show the results in Suppl data, viz.,  the high expression of EGFR in lung cancer cell lines, the effects of doses (5 and 10 mg/kg/d) in nude mice model. What does it mean about the” sensitivity of the drug against nude mice”, please specify? There are still some typo and grammar errors such as line 166 (slightly) and Western blot analysis (Western is the name of researcher, it (W) should be in capital letter. In Figure 4, ## mark should be in superscript form.

Author Response

  1. In each experiment there should be at least one reference included, please check and add the reference in all methods.

A: Thank you for your comments. As you pointed out, we have added the references in all methods.

  1. For the Western blotting bands, some contains 2 lanes and some more than 2, there should be labeling above them to mark each lane what it represents for.

A: Thank you for your comments. As you pointed out, we have revised the labeling in Figure 4, which is consistent to the Figure 3.

  1. For In Western blotting why did some lanes contain several bands and not consistent with other repeats. It seems that the immunoblotting results are not consistent with each other.

A: Thank you for your comments. Your comments are valuable. In fact, for economic, antibody was recycle used for more than two experiments. When the recycle antibody was used, it appeared several bands and not consistent with other repeats.

  1. For the data (that were not shown) it would be better to show the results in Suppl data, viz., the high expression of EGFR in lung cancer cell lines, the effects of doses (5 and 10 mg/kg/d) in nude mice model.

A: Thank you for your comments. Your comments are valuable. This study focused on the effect of FGFR inhibitor, then we assessed the expressions of FGFR 1-4 in different cell lines. As you pointed out, we have added the Suppl data for data not shown in manuscript.

  1. What does it mean about the” sensitivity of the drug against nude mice”, please specify?

A: Thank you for your comments. In fact, we mean that intraperitoneal injection of erdafitinib (10 mg/kg/day) in A549 xenograft mice for 21 days was more effective than erdafitinib (5 mg/kg/day) (Figure S4).

  1. There are still some typo and grammar errors such as line 166 (slightly) and Western blot analysis (Western is the name of researcher, it (W) should be in capital letter.

A: Thank you for your comments. Sorry for our negligence, we have revised it as you pointed out.

  1. In Figure 4, ## mark should be in superscript form.

A: Thank you for your comments. Sorry for our negligence, we have revised it as you pointed out.

Reviewer 2 Report

This reviewer could recommend this article except for one concern below.

(1)

Such negative results about p21 also should be addressed in the Discussion section.

(2) and (3)

This reviewer can accept the authors’ response. 

Author Response

  1. Such negative results about p21 also should be addressed in the Discussion section.

A: Thank you for your comments. Your comments are valuable. We have added the information of p21 in Discussion.

Round 3

Reviewer 1 Report

The manuscript is in the revised in the corrected form and is ready for publication.
